# Molecular Hydrogen Neuroprotection in Post-Ischemic Neurodegeneration in the Form of Alzheimer’s Disease Proteinopathy: Underlying Mechanisms and Potential for Clinical Implementation—Fantasy or Reality?

**DOI:** 10.3390/ijms23126591

**Published:** 2022-06-13

**Authors:** Ryszard Pluta, Sławomir Januszewski, Stanisław J. Czuczwar

**Affiliations:** 1Laboratory of Ischemic and Neurodegenerative Brain Research, Mossakowski Medical Research Institute, Polish Academy of Sciences, 02-106 Warsaw, Poland; sjanuszewski@imdik.pan.pl; 2Department of Pathophysiology, Medical University of Lublin, 20-090 Lublin, Poland; stanislaw.czuczwar@umlub.pl

**Keywords:** brain ischemia, neurodegeneration, amyloid, tau protein, dementia, molecular hydrogen, neuroprotection

## Abstract

Currently, there is a lot of public interest in naturally occurring substances with medicinal properties that are minimally toxic, readily available and have an impact on health. Over the past decade, molecular hydrogen has gained the attention of both preclinical and clinical researchers. The death of pyramidal neurons in especially the CA1 area of the hippocampus, increased permeability of the blood-brain barrier, neuroinflammation, amyloid accumulation, tau protein dysfunction, brain atrophy, cognitive deficits and dementia are considered an integral part of the phenomena occurring during brain neurodegeneration after ischemia. This review focuses on assessing the current state of knowledge about the neuroprotective effects of molecular hydrogen following ischemic brain injury. Recent studies in animal models of focal or global cerebral ischemia and cerebral ischemia in humans suggest that hydrogen has pleiotropic neuroprotective properties. One potential mechanism explaining some of the general health benefits of using hydrogen is that it may prevent aging-related changes in cellular proteins such as amyloid and tau protein. We also present evidence that, following ischemia, hydrogen improves cognitive and neurological deficits and prevents or delays the onset of neurodegenerative changes in the brain. The available evidence suggests that molecular hydrogen has neuroprotective properties and may be a new therapeutic agent in the treatment of neurodegenerative diseases such as neurodegeneration following cerebral ischemia with progressive dementia. We also present the experimental and clinical evidence for the efficacy and safety of hydrogen use after cerebral ischemia. The therapeutic benefits of gas therapy open up new promising directions in breaking the translational barrier in the treatment of ischemic stroke.

## 1. Introduction

Molecular hydrogen is an odorless, colorless gas that is physiologically inert. Hydrogen is the lightest and most abundant element in the Earth’s atmosphere. Furthermore, hydrogen is a biological gas that is produced in mammals by intestinal bacteria [1,2,3]. It is believed to be a new type of natural antioxidant with a low ability to react with most biomolecules, which is potentially therapeutic. The first use of hydrogen in humans was hydreliox, a breathing gas mixture of hydrogen, helium and oxygen that is used to prevent decompression sickness and nitrogen narcosis during very deep technical dives [4]. The therapeutic use of hydrogen was first demonstrated in 1975, showing that hyperbaric hydrogen caused marked regression of skin squamous carcinoma in mice [5]. In 2001, hydrogen was documented to have anti-inflammatory properties in experimental parasitic hepatitis [6]. Six years later, it was found that hydrogen was supposed to act as an antioxidant by selectively removing the neurotoxic hydroxyl radical and peroxynitrite from the parenchyma of the rat brain as a result of oxidative stress caused by ischemic brain injury [7]. Hydrogen has been shown to have many advantages as a neuroprotective gas. First, it can penetrate biomembranes, diffuse into the cytosol and organelles and cross the blood-brain barrier [7,8,9]. Secondly, the repeated administration of hydrogen does not cause tolerance [10]. Thirdly, different easy and convenient approaches to administer it are available [11,12]. Fourthly, hydrogen has a protective effect against many diseases, including peripheral and central nervous system diseases such as neuropathic pain, Alzheimer’s disease, stroke, animal cerebral ischemia and neonatal hypoxic-ischemic brain injury [11,13,14,15,16,17,18,19,20,21,22,23]. Fifth, there are no documented serious side effects [13,24]. Sixth, in Japan, 2% molecular hydrogen inhalation has been approved for the clinical treatment of cerebral ischemia due to cardiac arrest [24,25,26,27,28]. The Chinese National Health and Medical Commission in 2020 recommended the use of inhaled hydrogen in addition to oxygen therapy for anti-cancer, anti-inflammatory and anti-oxidant treatments [29]. Hydrogen has been suggested as a new complementary therapy against stroke, which, e.g., reduces oxidative stress, neuroinflammation and apoptosis [22,30,31,32]. Despite many inaccuracies, the selective ability to scavenge free radicals and heal inflammation are still widely accepted mechanisms of the action of hydrogen [32]. Clinical trials have shown that hydrogen treatment is safe and effective in patients with asthma and chronic obstructive pulmonary disease [33,34].

Recently, it has been proposed to prevent and treat coronavirus disease by inhaling oxygen mixed with hydrogen (33.4% oxygen and 66.6% hydrogen) due to the important role of hydrogen in alleviating the worsening of lung function, emphysema, and acute or chronic inflammation [29,35,36]. Thus, molecular hydrogen is an interesting potential therapeutic gas for the prevention and treatment of various diseases, including neurological disorders. In this way, hydrogen contributes to the further development of the healthcare industry by using its healing properties. In this review, we summarized the current understanding of the potential neuroprotective properties of molecular hydrogen towards global and focal cerebral ischemia and the possible molecular mechanisms associated with its beneficial activity.

## 2. Search Criteria and Data Collection

Published scientific papers on the use of molecular hydrogen have been screened for in vitro, in vivo, animal and clinical investigations and side effects. Searches were performed digitally using databases including PubMed, MEDLINE, SCOPUS, Google Scholar and Science Direct to identify peer-reviewed original articles and reviews over the past twenty years (1 January 2001–31 December 2021). The search strategy was carried out using the following key words: “Molecular hydrogen therapy and brain ischemia”, “brain ischemia and molecular hydrogen therapy”, “molecular hydrogen therapy and ischemic stroke”, “ischemic stroke and molecular hydrogen therapy”, “molecular hydrogen therapy and cardiac arrest”, “cardiac arrest and molecular hydrogen therapy”, “molecular hydrogen neuroprotection and brain ischemia”, “molecular hydrogen neuroprotection and ischemic stroke”, “ischemic stroke and molecular hydrogen neuroprotection”, “molecular hydrogen therapy and amyloid”, “molecular hydrogen therapy and tau protein”, “tau protein and molecular hydrogen therapy“, “molecular hydrogen and bioavailability”, “molecular hydrogen and side effects”. The work covered by the study had to be related to the search terms and be as up-to-date as possible. The excluded works did not include hydrogen compounds, including hydrogen sulfide and numerous works by one author or from the same laboratory, in this case the latest works were taken. A total of 575 original papers and reviews were found, and 109 publications closely related to the subject of the review were used.

## 3. Molecular Hydrogen Neuroprotection in Post-Ischemic Brain Injury

It is now known that neurodegeneration following cerebral ischemia is caused by numerous proteomic and genomic changes that lead to neuronal death by necrosis and apoptosis, with progressive brain neuroinflammation and atrophy, ultimately leading to full-blown dementia [37,38,39,40,41,42,43,44,45,46,47,48,49]. Research indicates that, after cerebral ischemia, neurodegeneration of the Alzheimer’s disease type develops with selective neuronal death in the hippocampus with its complete atrophy [50,51,52,53,54,55]. The development of neuroinflammatory lesions has been shown to play a key role in the progression of post-ischemic brain neurodegeneration [39,44,48]. Amyloid processing, tau protein modification, autophagy and mitophagy genes are involved in post-ischemic neurodegeneration in the same way as in Alzheimer’s disease [40,42,49,56,57,58,59,60,61,62,63,64,65,66,67].

Restoration of blood flow in the brain following cerebral ischemia and ischemic stroke triggers an outbreak of reactive oxygen species, triggering a neuroinflammatory response and oxidative damage [39,44,48,65]. Reactive oxygen species destroy the membranes of neuronal and neuroglial cells, inducing lipid peroxidation, so antioxidants come into play as a therapeutic option [65]. Molecular hydrogen has been recognized as an antioxidant that can buffer the destructive effects of oxidative stress in the brain following ischemia by selectively reducing cytotoxic reactive oxygen species [7,22,30]. 

### 3.1. In Animals

In mice with focal cerebral ischemia with reperfusion, molecular hydrogen significantly increases SOD and GSH-Px activity, reduces malondialdehyde levels and infarct volume, relieves cerebral edema, improves neurological outcomes and alleviates cognitive deficits (Table 1) [19,68,69]. In global cerebral ischemia caused by cardiac arrest in rats, hydrogen inhalation improves neurological outcomes, cognitive deficits and survival [20,70]. Hydrogen injection or inhalation after global cerebral ischemia due to cardiac arrest effectively controls neuronal death and microglia activation in the hippocampus and lowers serum levels of S100b protein (Table 1) [8,31,71]. Inhalation of hydrogen or in combination with hypothermia has been shown to be superior to hypothermia alone in global cerebral ischemia from cardiac arrest in rats [31,71,72]. Molecular hydrogen has been shown to protect the permeability of the blood-brain barrier after focal and global cerebral ischemia (Table 1) [8,9]. Hydrogen has been shown to protect against oxidative stress, neuroinflammation, and prevents the ischemic site from turning into a hemorrhagic focus in rats with local cerebral ischemia (Table 1) [73]. Additionally, it has been shown that the intraperitoneal injection of hydrogen-rich saline has healing properties after transient global cerebral ischemia in rats [74]. Since most of the damage in the above model occurs between 6 and 24 h after ischemia, the effective hydrogen protection period was much less than the 6 h of recirculation, so the protective effect of hydrogen-rich saline in this situation is quite limited [74]. Hydrogen treatment of mice after bilateral closure of the common carotid artery improves cognitive abilities and induces anti-apoptotic and antioxidant effects [75]. In rats, up to 7 days after middle cerebral artery occlusion with reperfusion and administration of hydrogen, a reduction in infarct volume, ischemic penumbra hyperperfusion, neurological and behavioral disorders and white matter damage were observed (Table 1) [76]. It was documented that hydrogen therapy significantly improved the 7-day survival rate of mice after global brain ischemia, from 8.3 to 50% [77]. Histopathological analysis revealed that hydrogen therapy significantly attenuated neuronal injury and autophagy in the hippocampal CA1 area and also brain edema, after 24 h of reperfusion [77]. The beneficial effects of hydrogen therapy on post-ischemic brain injury were associated with significantly lower levels of oxidative stress markers: malondialdehyde and 8-hydroxy-2′-deoxyguanosine in the brain parenchyma [77]. Hydrogen inhalation following 10-min transient global cerebral ischemia in rats that survived 3 days attenuated cognitive impairment [78]. This neuroprotective effect was associated with decreased pyramidal neuronal death in the CA1 region of the hippocampus and inhibition of oxidative stress [78]. Hydrogen inhalation improved survival and neurological deficit after global ischemia caused by cardiac arrest in rats [31]. It also prevented the increase in left ventricular end-diastolic pressure and the increase in serum IL-6 levels, and reduced mortality [31,79,80]. Hydrogen treatment increased the level of interleukin-10, vascular endothelial growth factor and leptin [81]. In addition, a reduction in mortality in rats after cardiac arrest and an effect on the restoration of the bioelectrical activity of the brain was noted [81]. The survival rate at 4 h was 78% in the hydrogen group and 22% in the placebo group [81]. In another global model of cerebral ischemia due to cardiac arrest in rats, increased survival and inhibition of autophagy were observed [82]. Hydrogen inhalation for 4 days improves neurological outcomes and survival after global cerebral ischemia due to cardiac arrest in systemic hypertension rats and is superior to treatment with mild hypothermia [71]. The intraperitoneal injection of hydrogen into rabbits in cardiac arrest improved 3-day survival and neurological deficits, reduced neuronal damage, and inhibited neuronal apoptosis [83]. The intraperitoneal injection of hydrogen decreased the indicators of oxidative stress in the blood and the parenchyma of the hippocampus and increased the activity of the antioxidant enzyme [83]. Rats given hydrogen-rich water before and after occlusion of the middle cerebral artery, surviving up to 14 days after focal ischemia, showed reduced infarct volume and improved neurological outcomes [84]. In addition, hydrogen prevented ischemia-induced decreases in parvalbumin and hypocalcin, and also reduced neuronal cell death induced by toxic glutamate [84]. In addition, hydrogen lowered the increased levels of intracellular Ca^2+^ caused by glutamate toxicity [84].

Subsequent studies found that rats that underwent global cerebral ischemia and were treated with hydrogen-rich saline had milder neuronal injury and a limited number of irreversibly damaged neurons in the brain [15]. Expression of miR-210, miR21 and NF-κB in the ischemic hippocampus at 6, 24 and 96 h was significantly reduced in the hydrogen-treated group [15]. Moreover, the number of Tregs cells after cerebral ischemia treated with hydrogen increased on days one and four after reperfusion [15]. These results indicate that the recovery from global cerebral ischemia in rats dosed with hydrogen-rich saline is most likely associated with an upregulation of Treg cell numbers [15]. In this study, hydrogen significantly increased the number of surviving ischemic pyramidal neurons in the CA1 region of the hippocampus [15]. In addition, the neurobehavioral test confirmed that hydrogen reduced damage to the brain after ischemia [15]. This neuroprotective effect may be due to the extensive spread of hydrogen throughout the brain. Hydrogen has a good diffusion rate, can easily penetrate the blood-brain barrier and reach deep brain structures, and is also able to reach the site of injury before revascularization to remove toxic oxygen free radicals [85].

In behavioral studies of rats after focal cerebral ischemia, administration of lactulose, which induces endogenous hydrogen production in the intestine, resulted in higher neurological scores and shorter escape latency in the Morris test [86]. Morphological studies using 2,3,5-triphenyltetrazolium chloride showed a smaller infarct volume, Nissel staining showed relatively distinct and intact neuronal cells and TUNEL staining showed fewer apoptotic neurons [86]. In biochemical studies, lactulose decreased the content of malondialdehyde in the brain, the activity of caspase-3, the concentration of 3-nitrotyrosine and 8-hydroxy-2-deoxyguanosine and increased the activity of superoxide dismutase [86]. Orally administered lactulose activated expression of NF-E2 related factor 2 (Nrf2) in the brain [86]. The antibiotics suppressed the neuroprotective effects of lactulose by reducing hydrogen generation [86]. Lactulose administered intragastrically had a neuroprotective effect in post-ischemic brain injury in rats, which is attributed to the production of hydrogen via fermentation of lactulose by intestinal bacteria and activation of Nrf2 [86].

Studies that assessed the most effective timing of hydrogen administration after local cerebral ischemia in rats showed a post-ischemic time interval of up to 6 h during which significant reductions in infarct volume and brain edema were observed, and neurological outcomes improved [87]. At that time, after local cerebral ischemia, hydrogen decreased 8-hydroxyl-2′-deoxyguanosine (8-OHdG), reduced the content of malondialdehyde, interleukin-1β, tumor necrosis factor-α and suppressed caspase 3 activity [87]. These results indicate that hydrogen has a neuroprotective effect when administered during 6 h post-ischemia [87]. 

One study used hydrogen-saturated saline for focal cerebral ischemia in rats to test whether hydrogen-saturated saline reduces apoptosis of neuronal cells through the p38 MAPK-caspase-3 signaling pathway [88]. The obtained data showed that hydrogen reduced apoptotic neuronal cell death and infiltration of inflammatory cells in the brain cortex of rats post-ischemia [88]. In the hydrogen-treated group, there was a significant decrease in p38 MAPK protein expression compared to the untreated group [88]. It was concluded that hydrogen-rich saline could exert anti-apoptotic neuroprotective effects via the p38 MAPK signaling pathway [88]. It has also been found that, in acute post-ischemic brain injury in mice, hydrogen reduces the levels of Bax and TNFα and induces an anti-inflammatory response by regulating IL-2 and IL-10 [75].

### 3.2. In Humans

Studies in patients with acute focal cerebral ischemia have shown that administration of molecular hydrogen by inhalation or intravenous infusion is safe, hydrogen has been found in the blood, and there was no effect of hydrogen on physiological parameters [13,25,89]. Patients with local cerebral infarction who received hydrogen inhalation as part of their treatment had a reduced infarct size, improved neurological outcomes and the ability to perform daily activities compared to untreated patients (Table 2) [16]. In a human study of global cerebral ischemia due to cardiac arrest, inhalation of low concentration molecular hydrogen positively affected brain function without adverse events (Table 2) [25]. In Japan, a phase II clinical trial was conducted in patients with cerebral ischemia after cardiac arrest, finding a positive effect on neurological functioning and the safety of molecular hydrogen inhalation (Table 2) [25,27]. While no side effects of hydrogen have been detected in animal studies, potential side effects should still be investigated due to diarrhea reported by a small number of patients after receiving hydrogen [89].

## 4. Molecular Hydrogen versus Amyloid and Tau Protein Modification

It is now well known that ischemic neurodegeneration of the brain is caused by a set of proteomic and genetic changes that lead to the death of neuronal cells in an amyloid and tau protein dependent manner [40,46,47,56,57,58,59,63,64]. Amyloid causes oxidative stress with progressive neuroinflammation leading to brain atrophy and dementia [38,39,42,43,44,46,48,49,50,52,53,65]. Hard data show that cerebral ischemia in animals and humans leads to the production and accumulation of amyloid in the form of diffuse and senile plaques [37,53,95,96,97,98,99]. Direct evidence suggests that tau protein hyperphosphorylation leading to the development of neurofibrillary tangles also plays a role in the development of ischemic neurodegeneration of the brain, such as in Alzheimer’s disease [41,47,100,101]. The signaling mechanisms generated by amyloid and the tau protein following cerebral ischemia are believed to play a critical role in the development of irreversible neurodegeneration and dementia [37,41,47,53,57,58,59,63,100,101].

The amyloid protein precursor was found to be significantly lowered in the brain in a transgenic mouse model of Alzheimer’s disease after treatment with hydrogen-rich water [102]. In these animals, the level of β-secretase was also significantly lowered in the brain after administration of hydrogen-rich water [102]. The result of the above changes was a reduction in amyloid deposition in the CA3 area of the hippocampus [102]. Additionally, the decreased level of the soluble amyloid protein precursor α in this model was partially recovered by the administration of hydrogen-rich water, clearly confirming the neuroprotective effect of hydrogen [102]. Taken together, these results indicate that treatment with hydrogen-rich water prevents proteolysis of the amyloid protein precursor towards amyloid [102]. 

In the transgenic model of Alzheimer’s disease, intracerebral administration of Pd hydride nanoparticles (a highly charged hydrogen carrier) effectively removes hydroxyl radicals, reduces amyloid production and aggregation, alleviates mitochondrial dysfunction, reverses synaptic deficits and inhibits neuronal death [22]. In vitro, hydrogen treatment enhances the antioxidant system in human SK-N-MC neuroblastoma cells under the influence of amyloid-stimulated oxidative stress through the induction of AMPK and upregulation of the Sirt1-FoxO3a axis, which prevents mitochondrial dysfunction and the generation of reactive oxygen species, thereby ultimately maintaining cells survival [103].

The administration of hydrogen-rich water significantly inhibited the phosphorylation of the tau protein in Ser404 and Ser422 in the transgenic model of Alzheimer’s disease [102]. This resulted in a significant reduction in the number of neurofibrillary tangles in the CA3 region of the mouse hippocampus [102]. These results indicate that treatment with hydrogen-rich water can reduce pathological modifications of the tau protein, which prevents damage to neurons and synapses and memory deficits in the transgenic model of Alzheimer’s disease [102].

## 5. Molecular Hydrogen Bioavailability

Currently, there are several methods of molecular hydrogen administration, i.e., hydrogen inhalation, infusion of hydrogen-rich saline, hydrogen-rich water provided by various types of molecular hydrogen donors/suppliers, as well as functional micro/nanomaterials that increase the concentration of hydrogen administered and thus the effectiveness of treatment [12,104,105]. In the clinic, the most common methods of administering hydrogen are inhalation, infusion of hydrogen-rich saline and drinking hydrogen-rich water [28]. In clinical conditions, breathing hydrogen gas in concentrations of 1–4% is a hassle-free, direct and easy-to-use method [28]. It should be added that drinking hydrogen-rich water is safer and more convenient to use [28]. In contrast, hydrogen-rich saline is commonly used by intravenous infusion or intraperitoneal injection [28] Drinking hydrogen-rich water causes 59% of the consumed hydrogen to be expelled through the breath, almost 40% is used up in the human body, and about 1% is released through the skin [106]. It has been proven that after inhalation of 3% or 4% hydrogen, its level in venous and arterial blood rises rapidly and reaches a plateau within 20 min [13,24]. After stopping hydrogen inhalation, its level in arterial blood drops below 10% of the plateau level after around 6 min, and in venous blood after around 18 min [13,24]. However, after a 30-min intravenous infusion of hydrogen-rich saline in two volunteers, the levels of hydrogen in the venous and arterial blood rose to a plateau within 15 min and fell sharply after the infusion was stopped [13,24]. Data from clinical and experimental studies have shown that inhaling hydrogen produces higher concentrations of hydrogen in the brain tissue than other ways of its administration [13,24,107]. In order to select the most effective hydrogen treatment for a given disease entity, additional studies of the pharmacokinetics and medical benefits of combining hydrogen with water or other hydrogen carriers are necessary [28].

## 6. Conclusions

Molecular hydrogen exerts a significant neuroprotective effect on almost all animal models of cerebral ischemia studied. The neuroprotective properties of hydrogen are often documented by assessing neuronal survival and improved neurocognitive function (Figure 1). In addition, its protective properties against mitochondrial damage and the permeability of the blood-brain barrier were noted (Figure 1). Furthermore, studies showed that hydrogen-rich water prevented neuronal death and synaptic loss, inhibited the development of senile amyloid plaques and reduced tau protein hyperphosphorylation and the growth of neurofibrillary tangles in the transgenic model of Alzheimer’s disease (Figure 1). Few side effects indicate that the therapeutic potential of hydrogen is not limited and its neuroprotective effects are pleiotropic. Molecular mechanisms of hydrogen-initiated neuroprotection have been studied in animal focal and global cerebral ischemia demonstrating a reduction in oxidative stress, inhibition of pro-apoptotic effects with simultaneous activation of anti-apoptotic pathways and inhibition of the neuroinflammatory process with a simultaneous increase in the neurotrophic effect of microglia. It is well known that oxidative stress affects the expression of a large number of genes that stimulate many biological phenomena, such as increased production of amyloid, modification of the tau protein, autophagy, apoptosis and neuroinflammation, hence the antioxidant effect of hydrogen may be its most important neuroprotective property (Figure 1). Currently, two proposals for the molecular mechanisms underlying the antioxidant properties of hydrogen are put forward. A widely accepted theory is that hydrogen directly reacts with hydroxyl radicals and peroxynitrite, the traditional scavenger theory [28]. The second theory suggests that hydrogen controls the production of reactive oxygen species by acting as a rectifier for the electron flow in the mitochondria [28]. Although the neuroprotective effects of hydrogen have been demonstrated in the last decade in numerous experimental studies and after preliminary clinical trials, further work is needed in the clinical use of hydrogen. Research has shown that hydrogen is non-toxic but has minor side effects including heartburn, diarrhea and headache, which have been reported in a few cases [89,108]. The distribution of hydrogen in organs and tissues varies with the mode of administration, which impacts the medical benefit. It seems that the limited therapeutic effect of hydrogen may be related to the duration, concentration and method of application, beyond the phase of a specific disease entity. For these reasons, additional studies are required to study the pharmacokinetics and effect of hydrogen dosage and then develop more effective delivery regimens/procedures. Carefully completed translational studies in various models of cerebral ischemia in animals are needed to demonstrate whether the neuroprotective effects of hydrogen on post-experimental cerebral ischemia fully exist, thus paving the way for a design for clinical use.

## 7. Outlook

Currently, ischemic stroke is one of the most important causes of morbidity and mortality worldwide. At present, tissue plasminogen activator is administered to induce thrombolysis and restore cerebral blood flow following focal brain ischemia, but therapeutic benefit is only achieved in a small percentage of patients qualified for fibrinolysis. Despite the evident preclinical confirmation of the neuroprotective hydrogen treatment of local post-ischemic brain injury, it has not been successfully translated into clinical use. So, the treatment that will please most acute stroke patients remains a mystery. Currently, despite the lack of effective treatment of the consequences of ischemic stroke, research is being carried out on new methods of treating this disease, however, the use of medicinal gases, especially hydrogen, is not very popular. Nevertheless, it should be stated that gases such as normobaric and hyperbaric oxygen and hydrogen exert medical effects in preclinical studies of cerebral ischemia. There are significant advantages in using gases in terms of low cost, high quantity and ease of administration, all of these properties making them ideal candidates for the translational treatment of ischemic stroke. In conclusion, the influence of cellular gaseous mediators such as carbon monoxide, nitric oxide and hydrogen could be an interesting alternative for treating ischemic stroke [1,2,3]. Breathing with these gas mediators may additionally generate neuroprotection, but this plan must be validated as an effective treatment for stroke. This review reveals the neuroprotective potential of hydrogen treatment, supporting the possibility of modulating cellular gas mediators following cerebral ischemia. The benefits of hydrogen therapy open up new promising directions in breaking the translational barrier for brain ischemia in humans.

The evidence introduced in this review shows a promising neuroprotective effect of hydrogen after cerebral ischemia with recirculation (Figure 1). However, the limited number of experimental and clinical studies demonstrating the neuroprotective effects of hydrogen following cerebral ischemia does not provide sufficient scientific support. In addition, recirculation times following cerebral ischemia were short, and therefore the effects of hydrogen treatment related to long-term follow-up following ischemic injury are currently unknown. Therefore, future randomized clinical investigations are needed to confirm the effectiveness of hydrogen therapy and to provide data on some of the currently unresolved issues, such as the timing of hydrogen use. Despite very scarce evidence, the current research on hydrogen in the treatment of cerebral ischemia with reperfusion seems interesting and promising for the future as a neuroprotective molecule and preventing the deposition of various amyloid plaques and dysfunctional tau protein in the form of neurofibrillary tangles (Figure 1). In the last decade, the interest in hydrogen and its reputation for its numerous pharmacological effects has been steadily increasing. Due to the fact that hydrogen, like many other natural substances, has more than one drug target (Figure 1), it indicates its versatile use and low risk of treatment resistance. There is no doubt that, due to preclinical data, the next step in investigating the therapeutic properties of hydrogen must be well-designed and controlled randomized clinical trials. Double-blind research is imperative to finally elucidate the medical properties of hydrogen. A definitive elucidation of the therapeutic benefits of hydrogen may offer hope for the long-term effect of a therapy that is currently a feverish subject of research. Hydrogen is currently not approved for clinical use except in Japan and China [13,24,25,26,27,29]. High bioavailability and negligible side effects of hydrogen administration are an invaluable positive element of its usefulness in the clinic. We hope that future clinical research will help us better understand the medical potential of hydrogen and place this fascinating gas at the forefront of new neuroprotective therapies in neurodegeneration.

## Figures and Tables

**Figure 1 ijms-23-06591-f001:**
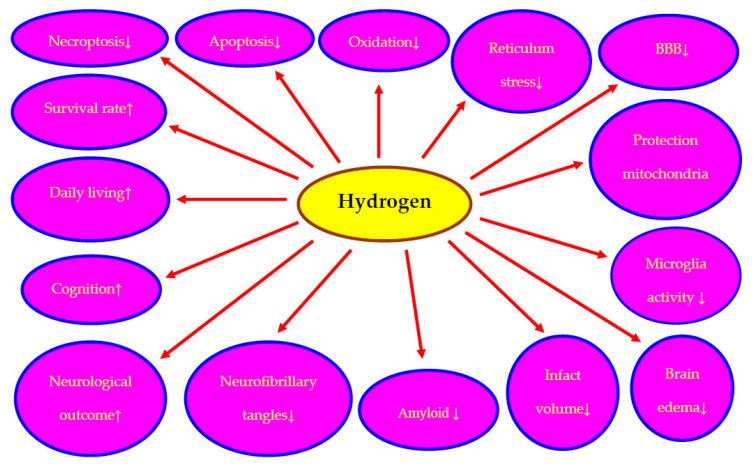
Neuroprotective mechanisms of molecular hydrogen on post-ischemic brain neurodegeneration phenomena. BBB-blood-brain barrier. ↓—reduction, ↑—increase.

**Table 1 ijms-23-06591-t001:** Neuroprotective effects of molecular hydrogen in experimental post-ischemic brain injury.

Ischemia	Animal	Strain	Treatment	Benefits	References
Focal	Mice	C57B/L	Inhalation of 66.7% hydrogen/33.3% oxygen for 90 min post-ischemia.	Inhibition of microglial activity and regulation of microglial phenotype. Improvement of neurological outcome.	[19,68]
Global	Mice	C57BL/6J	Inhalation hydrogen (1.3%), oxygen (30%), and nitrogen (68.7%). 45 min of ischemia and 180 min of reperfusion, and 3 h/d, from 1 to 3 days post-ischemia.	Improved survival. Attenuation of neuronal injury, autophagy and brain edema.	[77]
Global	Rat	Wistar	2.1% hydrogen supplemented by room air ventilation for 4 h after ischemia.	Reduction changes of prooxidant enzyme and gap junction protein levels.	[90]
Global	Rat	Sprague-Dawley	Hydrogen-rich saline (5 mL/kg) was injected immediately post-ischemia.	Significant improvement of surviving cells. Reduction tissue damage, the degree of mitochondrial swelling, and the loss of mitochondrial membrane potential but also preservation the mitochondrial cytochrome c content.	[91]
Global	Rat	Sprague-Dawley	I.V. hydrogen-rich saline (1 mL/kg, 4 mL/kg, or 6 mL/kg), HRS was given before hypoxia and during reoxygenation.	Inhibition of hippocampus endoplasmic reticulum stress and microvascular endothelial cells apoptosis via PI3K/Akt/GSK3β signaling pathway.	[92]
Global	Rat	Sprague-Dawley	Hydrogen-rich saline 5 mL/kg was intraperitoneally injected immediately and 6 h post-ischemia.	Significant improvement survival rate and neurological function. The beneficial effects associated with decreased levels of oxidative products, as well as the increased levels of antioxidant enzymes and accompanied by the increased activity of glucose-regulated protein 78, the decreased activity of cysteinyl aspartate specific proteinase-12 (caspase-12).	[93]
Global	Rat	Wistar	Inhalation of 2% hydrogen started immediately at the end of ischemia and lasted for 3 h.	Attenuation of cognitive impairment. Decreased pyramidal neuronal death in CA1 region of hippocampus.	[78]
Global	Rat	Sprague-Dawley	Hydrogen-rich saline was administered i.v. at 1 min before end of ischemia, followed by injections at 6 and 12 h post-ischemia.	Improves survival and neurological outcome.	[8]
Focal	Rat	Sprague-Dawley	6 mL/kg i.p. per rat before and after ischemia.	Reduction brain infarct volume and improvement of neurological function. Prevention the ischemia-induced reduction of parvalbumin and hippocalcin levels and also reduced the glutamate toxicity-induced death of neurons. Attenuation the glutamate toxicity-induced by elevate in intracellular calcium.	[84]
Focal	Rat	Sprague-Dawley	0.5 mL/kg/day saturated hydrogen saline (0.6 mmol/L) i.p. 3 days prior to ischemia and immediately during 24 h of reperfusion.	Significantly reduction the number of apoptotic cells, and the protein expression of p38 MAPK and caspase-3. These effects may be associated with the p38MAPK signaling pathway.	[88]
Focal	Rat	Sprague-Dawley	Hydrogen saline was injected i.p. (1 mL/100 g body weight) at designed time points 0, 3 or 6 h after reperfusion onset.	Reduction 8-hydroxyl-2′-deoxyguanosine, malondidehyde, interleukin-1β, tumor necrosis factor-α, and suppressed caspase 3 activity in ischemic brain.	[87]
Global	Rabbit	White	Before ischemia i.p. injection of hydrogen low dose (10 mL/kg) or high dose (20 mL/kg).	Improvement survival and neurological outcomes, reduction of neuronal damage and inhibition of neuronal apoptosis. Reduction indicators of oxidative stress in the blood and the hippocampus and increased activity of antioxidant enzyme.	[83]
Global	Swine	Yorkshire	Inhalation of hydrogen (2.40%) for a 24-h period during and after the ischemic injury.	Reduced neurological injury.	[94]

**Table 2 ijms-23-06591-t002:** Neuroprotective effects of molecular hydrogen in clinical post-ischemic brain injury.

Ischemia	Number of Participants	Treatment	Benefits	Study	References
Focal	50 patients	Inhalation 3% hydrogen gas (1 h twice a day) for initial 7 days.	Reduced infarct size, improved neurological outcome and daily living activity.	Randomized	[16]
Global	5 patients	2% hydrogen with oxygen was supplied via a respirator after admission to the intensive care unit for 18 h.	4 patients survived 90 days with a favorable neurological outcome.	Pilot study	[25]
Global	360 patients	2% hydrogen with 24 to 50% oxygen was supplied via mechanical ventilation after admission for 18 h.	The first multicenter randomized trial is underway to confirm the efficacy of hydrogen on neurological outcomes in comatose out-of-hospital cardiac arrest survivors.	Randomized, double-blind, placebo-controlled trial.	[26]
Global	5 patients	Inhalation 2% hydrogen with titrated oxygen was initiated upon admission for 18 h.	Oxidative stress markers were reduced in cardiogenic post-cardiac arrest patients but were slightly elevated in the patient with sepsis. Inflammatory cytokine levels remained unchanged in cardiogenic post-cardiac arrest patients, whereas a dramatic reduction was observed in one patient with sepsis.	Pilot study	[27]

## Data Availability

The data is available from the author for correspondence.

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
