# Peer review of "Molecular Hydrogen Neuroprotection in Post-Ischemic Neurodegeneration in the Form of Alzheimer’s Disease Proteinopathy: Underlying Mechanisms and Potential for Clinical Implementation—Fantasy or Reality?"

_ijms, 2022, doi:10.3390/ijms23126591_

Round 1
Reviewer 1 Report
In my opinion, the submitted review is a well-written and very informative one. The authors presented the current knowledge on the potential neuroprotective properties of molecular hydrogen towards cerebral ischemia concisely but clearly and comprehensively.
I have only two minor suggestions:
1. In the section “Search and data collection” inclusion and exclusion criteria should be briefly mentioned. The expression: “closely related to the subject of the review” is too enigmatic.
2. I suggest dividing the Table 1 into two separate tables: the first table with findings from preclinical studies and the second table with findings from clinical trials.
In the first table with findings from preclinical studies, two additional columns should be added: Animal strain and Treatment schedule (duration, dosage/concentration, method of application).
In the second table with findings from clinical trials, apart from the type of ischemia, benefits, and references, information about the number of participants, treatment schedule (duration, dosage/concentration, method of application), and whether a given study was controlled, randomized and blinded should be added.
Author Response
Reviewer 1. In my opinion, the submitted review is a well-written and very informative one. The authors presented the current knowledge on the potential neuroprotective properties of molecular hydrogen towards cerebral ischemia concisely but clearly and comprehensively. I have only two minor suggestions: 1. In the section “Search and data collection” inclusion and exclusion criteria should be briefly mentioned. The expression: “closely related to the subject of the review” is too enigmatic. Done. 2. I suggest dividing the Table 1 into two separate tables: the first table with findings from preclinical studies and the second table with findings from clinical trials. In the first table with findings from preclinical studies, two additional columns should be added: Animal strain and Treatment schedule (duration, dosage/concentration, method of application). In the second table with findings from clinical trials, apart from the type of ischemia, benefits, and references, information about the number of participants, treatment schedule (duration, dosage/concentration, method of application), and whether a given study was controlled, randomized and blinded should be added. Done.
Reviewer 2 Report
Hydrogen therapy has attracted increasing attention in clinical/preclinical studies as an emerging and promising therapy against several diseases due to its pleiotropic neuroprotective properties. In the current review the authors refer to current and future therapeutic challenges in post ischemic neurodegeneration highlighting the promising therapeutic options of hydrogen. The study is well written, well organized and the table very helpful.
Comments
1)Title: While in the title the authors refer to Alzheimer disease, in the abstract they mention that ‘This review focuses on assessing the current state of knowledge about the neuroprotective effects of molecular hydrogen following ischemic brain injury’ without specific reference to Alzheimer’s disease (AD). Furthermore in the text the reference to Alzheimer’s disease is very limited while the authors often mention tau protein and amyloid, but tau protein and amyloid abundance or plaques are also found in other neurodegenerative diseases, except for AD. Thus, the title should be changed accordingly.
2) References: Some references need to be corrected according to the journal instructions (pages, etc)
Author Response
Reviewer 2. Hydrogen therapy has attracted increasing attention in clinical/preclinical studies as an emerging and promising therapy against several diseases due to its pleiotropic neuroprotective properties. In the current review the authors refer to current and future therapeutic challenges in post ischemic neurodegeneration highlighting the promising therapeutic options of hydrogen. The study is well written, well organized and the table very helpful. Comments 1)Title: While in the title the authors refer to Alzheimer disease, in the abstract they mention that ‘This review focuses on assessing the current state of knowledge about the neuroprotective effects of molecular hydrogen following ischemic brain injury’ without specific reference to Alzheimer’s disease (AD). Furthermore in the text the reference to Alzheimer’s disease is very limited while the authors often mention tau protein and amyloid, but tau protein and amyloid abundance or plaques are also found in other neurodegenerative diseases, except for AD. Thus, the title should be changed accordingly. 2) References: Some references need to be corrected according to the journal instructions (pages, etc) It should be emphasized that the work does not concern Alzheimer's disease, but cerebral ischemia, which has similar or identical changes as included in the title to Alzheimer's disease.
Reviewer 3 Report
Paper is well written, structure is suitable to cover all aspects of the newly emerging topic of molecular hydrogen usage in experimental and clinical medicine. Enlargement and more detailed description of the possible molecular pathways linked with the hydrogen application can potentially increase the readability and scientific soundness of the paper.
Author Response
Reviewer 3. Paper is well written, structure is suitable to cover all aspects of the newly emerging topic of molecular hydrogen usage in experimental and clinical medicine. Enlargement and more detailed description of the possible molecular pathways linked with the hydrogen application can potentially increase the readability and scientific soundness of the paper. We present in this paper all the mechanisms of hydrogen neuroprotective action available today (descriptions, tables and figure). There are no new data as of today in the literature.